# Design of an Energy Policy for the Decarbonisation of Residential and Service Buildings in Northern Portugal

Sara Capelo [1], Tiago Soares [2,*], Isabel Azevedo [3], Wellington Fonseca [1,2] and Manuel A. Matos [1,2]

1  Faculty of Engineering, University of Porto, 4200-465 Porto, Portugal
2  Center for Power and Energy Systems, Institute for Systems and Computer Engineering,
   Technology and Science (INESC TEC), 4200-465 Porto, Portugal
3  Institute of Science and Innovation in Mechanical and Industrial Engineering (INEGI),
   4200-465 Porto, Portugal
*  Correspondence: tiago.a.soares@inesctec.pt

**Abstract:** The decarbonisation of the building sector is crucial for Portugal's goal of achieving economy-wide carbon neutrality by 2050. To mobilize communities towards energy efficiency measures, it is important to understand the primary drivers and barriers that must be overcome through policymaking. This paper aims to review existing Energy Policies and Actions (EPA) in Portugal and assess their effectiveness in improving Energy Efficiency (EE) and reducing $CO_2$ emissions in the building sector. The Local Energy Planning Assistant (LEPA) tool was used to model, test, validate and compare the implementation of current and alternative EPAs in the North of Portugal, including the national EE plan. The results indicate that electrification of heating and cooling, EE measures, and the proliferation of Renewable Energy Sources (RES) are crucial for achieving climate neutrality. The study found that the modelling of alternative EPAs can be improved to reduce investment costs and increase Greenhouse Gas (GHG) emissions reduction. Among the alternatives assessed, the proposed one (Alternative 4) presents the best returns on investment in terms of cost savings and emissions reduction. It allows for 52% investment cost savings in the residential sector and 13% in the service sector when compared to the current national roadmap to carbon neutrality (Alternative 2). The estimated emission reduction in 2050 for Alternative 4 is 0.64% for the residential sector and 3.2% for the service sector when compared to Alternative 2.

**Keywords:** building energy consumption; energy efficiency; energy policies and actions; greenhouse gas emissions; local energy planning; net zero carbon building

## 1. Introduction

### 1.1. Background and Motivation

In the last couple of decades, EPA has been increasingly important in improving the EE of buildings in Europe, allowing citizens to reduce energy costs whilst improving comfort. Portugal was one of the first nations in the world to set goals for achieving carbon neutrality by 2050 [1]. The main methods used by Portugal's energy and climate policies to achieve carbon neutrality are enhanced EE, the rapid expansion of renewable electricity generation, and significant electrification of energy demand. The primary concerns are reducing reliance on energy imports and keeping access to inexpensive energy [2]. Currently, there are several EPA for the empowerment of consumers through energy communities to improve EE levels, reduce energy consumption and increase the levels of self-consumption and self-sufficiency in cities.

Community action on energy has increased significantly over the past decade, spurred by concerns about climate change and rising energy costs [3]. Energy Community (EnC) is a promising framework to enable a decarbonised economy. However, it is yet to be well understood by the overall population. EnC initiatives are more likely adopted by citizens who prefer to be actively involved in environmental issues. EnC measures often

require upfront investments, and their relationship to long-term individual benefits is non-trivial to the general public. Thus, it is of utmost importance to mobilize communities, as well as provide accessible information, to support and actively participate in initiatives that promote energy reduction or the production of energy from RES. Although, the most widely identified barrier is the suite of financial constraints holding back community action [3]. This means that the government must continue to improve and guarantee investment funds that support citizens and increase consumer mobilization to participate in energy communities.

This work aims to map the evolution of EPA towards the improvement of buildings' energy performance in Portugal. Based on this review, it is also intended to assess the impact of existing buildings-related energy policy actions in the North of Portugal in terms of required investment and GHG emissions, as well as to propose alternative scenarios.

### 1.2. Literature Review

Energy use is a subject of the utmost relevance due to the urgency of sustainability and its intrinsic relationship with society and quality of life. Over the last decades, it has been thoroughly discussed about anthropic driven climate change. In 2016, Paris Agreement set worldwide goals towards decarbonisation by 2050 and keeping temperature increase below the 1.5 °C threshold [4]. The carbon-intensive society, as observed after the industrial revolution, is non-compliant with the maintenance of global temperature levels. Therefore, achieving net-zero carbon emissions is imperative. According to [5], efficiency improvement plans and increasing usage of RES are among the crucial steps to be taken toward GHG emissions mitigation. More than a third of the world's final energy consumption and over 40% of all pollutant emissions are attributed to the buildings sector, making it a crucial target for EE and resource conservation policies. In Portugal, the building sector represents 30% of final consumption [6]. The Energy Performance of Buildings Directive (EPBD) is a European policy aimed at improving the energy efficiency of buildings, encompassing both the building envelope and the respective systems, such as water heating and space heating and cooling.

The literature is rich in reviewing the different aspects of the EPA, as well as in simulating and analyzing its impact in different countries, especially at the European Union (EU) level [1]. However, it often overlooks its impact and suitability at the local governance level. In this particular, the authors in [7] examine the lessons discovered when developing national EE programs and trace the development from 2007 to 2020. The adoption of standardised reporting ways, the installation of stronger monitoring systems, the development of evidence-based target-setting methodologies, and a wider consideration of policy packages resulted in significant improvements in the most recent national energy and climate plans. The need to set up systems that encourage the adoption of targets in keeping with a country's cost-effective EE potential is one of several areas with substantial potential for additional improvements, along with the need for a more cohesive reporting structure for policies and initiatives.

Nevertheless, a few works have sought the suitability and applicability of the national EPA in Portugal. Using the guidelines of the Portuguese Government's objectives for the following decades and a significant portion of renewable energy supply, the work in [8] presents technical alternatives on a Portuguese scale for the power system to achieve the target of reducing $CO_2$ emissions. The study shows the significance of pumping hydro-power plants for the integration of variable RES and establishes the least load capacity value of thermal power plants necessary to sustain the security standards of the Portuguese electrical system. The results demonstrated that RES could completely meet Portugal's long-term electricity needs in terms of energy balance. Ferreira et al. [9] analyzes the Portuguese Energy Certification System (ECS), and the regulation applied to the residential sector, considering complementary solutions for improving EE in residential buildings. The proposed solutions include interior and exterior thermal insulation, a lower air renovation ratio, light-coloured exterior walls, a better glazing system, shading elements, and a

Trombe Wall. The latter provides both convective and radiative heat within a household. Complementary, the LNEG [10] developed software to calculate the results of applying the ECS 2006 methodology. Then, the PTnZEB [11] calculation tool, currently employed by qualified experts, was developed for implementing the methodology of roadmap 2013 and 2016. The approaches were tested and analyzed on a single-family home that was constructed and approved in 2009, as detailed in [12]. Indicators assessed include energy needs, final and primary energy consumption, $CO_2$ emissions, and share of RES. Although having the same end goal of promoting domestic EE, each member state of the EU employed different measures according to their specific needs. For instance, to decrease energy use, some nations have implemented building performance requirements, information campaigns, and building energy codes, among other measures. Others recommended using financial tools like tax rebates and subsidies [13]. In this way, Ref. [13] assessed the effects of financial and tax incentive policies on the EE of residential properties in 19 districts in Portugal from 2014 to 2021. The results suggested that buildings with higher EE (certificates A+, A, and B) are negatively impacted by per capita income. As a result, Portugal's income is insufficient and prevents investment in highly energy-efficient homes. In the province of Teruel, located in the Autonomous Community of Aragón in northeastern Spain, a detailed study was conducted to analyse the support to local authorities through technical and financial guidance in the development of tools for the implementation of energy efficiency measures [14]. The study concluded that access to precise building data, such as that regarding aspects of the building envelope, is one of the obstacles discovered. From the literature, it is clear that EU member states must employ energy policy instruments suited to their economic and energy structures in the buildings sector because adopting well-designed policies can result in significant energy savings [13].

Regarding the US, buildings are responsible for 31% of $CO_2$ emissions and 40% of energy use [15]. Existing building energy retrofits offer a practical way to lower carbon footprints and building use. Predicting how different prospective retrofits will affect energy use is a crucial element in the planning process. Currently, decision-makers rely on simulation-based technologies to do in-depth analyses of a wide range of retrofit choices. Simulators frequently need specialised knowledge, considerable computational capacity, and precise inputs for building characteristics, which makes it difficult to consider building portfolios or assess expansive policy ideas. In [15], the authors present a data-driven approach to generalising the heterogeneous treatment effect of past retrofits to forecast future savings potential for helping retrofit planning using data from a portfolio of 550 government buildings. However, for achieving net zero GHG emissions in buildings, it is necessary to undergo a widespread economic transition. In this particular, deep decarbonization methods place a lot of emphasis on buildings because they are substantial energy consumers and sources of GHG emissions [16]. Previous initiatives to improve building energy efficiency show some of the opportunities and limitations. Although not as quickly as many observers anticipated, decades of energy efficiency initiatives have improved the energy performance of the building stock.

Nevertheless, current studies have been scarce in analyzing the impact of the current and future EPA in Portugal, more precisely at the local/regional level, for achieving ambitious carbon-neutral targets by 2050. Therefore, the present study highlights the impact of the current EPA in Portugal at the local level, namely in the North of Portugal.

*1.3. Main Contributions*

This work proposes to study in detail the current and future EPA for residential and service buildings being implemented in Northern Portugal. More precisely, it addresses the changes needed to achieve the ambitious carbon-neutral targets by 2050. Based on the National Energy and Climate Plan (PNEC) and the National Carbon Roadmap (RNC), a set of EPA scenarios is built (also called alternative scenarios). Note that the PNEC establishes specific lines of action to reduce carbon intensity and promote the energy renewal of the

building stock for 2030, while the RNC defines the roadmap to reach carbon neutrality by 2050.

Thus, with the objective of simulating the economic, environmental and social impacts of current and future EPA on residential and service buildings in northern Portugal, the main contributions of this paper are threefold:

- To provide a detailed review of the current energy policies applied in Portugal;
- To perform a detailed analysis of the impact of the current EPA on the decarbonisation of buildings in Northern Portugal, based on the national plans for the decarbonisation of buildings;
- To design, develop and simulate a new energy policy framework for carbon neutrality in buildings by 2050 with lower investment costs compared to current national plans.

### 1.4. Paper Structure

The remainder of the paper is structured as follows. Section 2 presents an overview of Portuguese EPA focused on renewable energy and EE in buildings. Section 3 describes the LEPA tool used for the design, simulation and analysis of the potential impact of EPA applied to a local and/or regional context. Section 4 describes the case study focused on the simulation of the current EPA in Portugal in the specific context of Northern Portugal. Finally, Section 5 gathers the most important conclusions.

## 2. Portuguese Energy Policies

### 2.1. Portuguese Policies and Actions on Renewable Energy

Renewable energy policies are intended to incorporate environmental benefits, particularly the reduction of GHG emissions compared to fossil fuel-based energy systems. In 2021, fossil fuel accounted for 65.4% of primary energy consumed in Portugal, followed by the energy used to produce electricity (14%), biomass (4.9%) and others (5.9%) [17]. Thus, there are still challenges ahead to fully decarbonise the energy system. The electricity sector, however, presents a significant share of RES, which accounted for 49.3% of total generation in 2022 [18].

According to the EU Renewable Energy Directive (2009/28/EC) [19], and the respective recast—EU Directive (2018/2001) [20], the EU established the goal of increasing the share of RES, while Portugal committed to the target of 31% share of energy from RES in the gross final energy consumption and 10% share in the transport sector by 2020 [21]. The most relevant national energy and climate policy instrument for 2021–2030 is Portugal's PNEC for the 2030 horizon. The PNEC 2030 sets the following national goals (Figure 1): (i) to reach 15% of electricity interconnections to encourage EU countries to interconnect their installed electricity production capacity; (ii) to incorporate 47% of energy from RES in the final gross energy consumption, and 10% in the transport sector; (iii) to achieve a reduction of 35% in primary energy consumption to improve EE; and (iv) to reduce GHG emissions by 45 to 55%, compared to emissions recorded in 2005 [21].

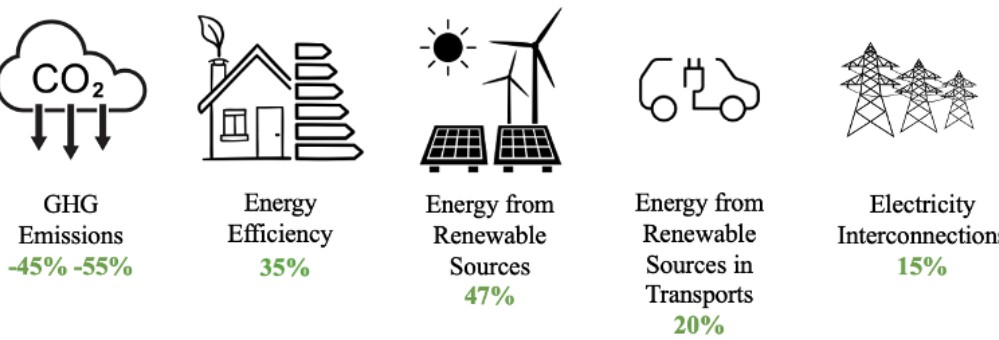

**Figure 1.** PNEC national targets for 2030, adapted with permission from Ref. [22].

This plan seeks to set targets and objectives on GHG emissions, renewable energy, EE, energy security, internal market, research, innovation, and competitiveness, along with a clear strategy for achieving them. Portugal met its challenge of incorporating 34.0% of RES in energy consumption by 2020. Therefore, it was essential to create new targets for 2021–2030 [22].

*2.2. Portuguese Policies and Actions on Energy Efficiency in Buildings*

EE can be defined as the optimization of energy consumption [23]. The threat of depletion of reserves of fossil fuels, the pressure of economic results, and environmental concerns are crucial points that define EE as one of the solutions to tackle climate change. In parallel, the buildings sector is responsible for the consumption of approximately 40% of final energy in Europe and around 30% in the case of Portugal. However, it is known that more than 50% of this consumption can be reduced by EE measures, which represents an annual reduction of 400 million tons of $CO_2$, almost the totality of the amount established in the Kyoto Protocol [24].

According to this, Portugal has assumed a path towards carbon neutrality. In this transition, priority was given to EE and the reduction of energy consumption, which will have energy sufficiency as a fundamental basis. Portugal had established the concern with energy consumption in buildings since the early 90s when the regulation of thermal behaviour characteristics of buildings and air-conditioning energy systems codes were published [12], due to the first regulatory base to guarantee thermal comfort and building quality. The adjustment of national legislation imposed by the EPBD published in the Directive (2002/91/EC) [25] settled that the member states of the EU had to implement an ECS to inform the citizens about the thermal quality of buildings when constructing, selling or renting them. This document also states that this certification should cover all residential and service buildings, public or private. Consequently, the base legislation of the new ECS, established by Decree-Law (DL) (118/2013) [26], has been revised several times since 2013. The various ordinances, dispatches and associated laws also underwent some changes. The Environmental Fund was published through the DL (42-A/2016) [27]. In 2018, Directive (2010/31/EU) [28] was replaced by Directive (2018/844/EU) [29], which consists of the EPBD recast. Hence, in 2020, was published the DL (101-D/2020) [30], which partially transposes the EPBD recast and establishes the requirements applicable to buildings to improve their energy performance and regulates the ECS for buildings. After that, in 2021, the DL (102/2021) [31] establishes the requirements for accessing and exercising the activity of technicians in the building ECS. In fact, in January of 2021, a new regulation was implemented in Portugal, which states that new buildings must be Nearly-Zero Energy Buildings (NZEB) [32]. It is up to each EU member state to define NZEB and establish the concept in their own legislation. Finally, to better interpret the timeline of these policies, (Figure 2) depicts a diagram explaining the evolution of EE policies for buildings in Portugal.

*2.3. The Portuguese RoadMap to Carbon Neutrality*

One of the goals of the RNC 2050 [33] is to visualise society in 2050 and the trajectory that will enable Portuguese society to attain the political goal of carbon neutrality in that year. The goal is to reduce emissions by 97% (heating) and 96% (cooling) in the residential sector and 100% in the service sector (compared to 2005 levels), with the incorporation of renewable energy in heating and cooling to be 66% and 68% [33]. Figures 3 and 4 illustrate the objectives that the roadmap (RNC [33]) sets out over the time horizon for the residential and service building sectors, respectively. One of the alternatives addressed in the present study (Alternative 2) is based on these targets.

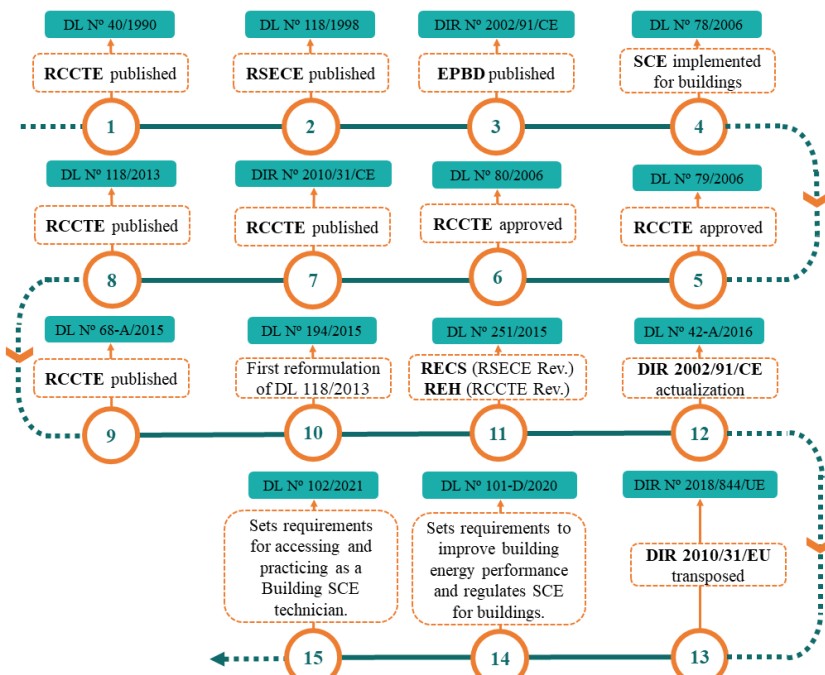

**Figure 2.** Timeline of Portugal's energy efficiency policies in buildings.

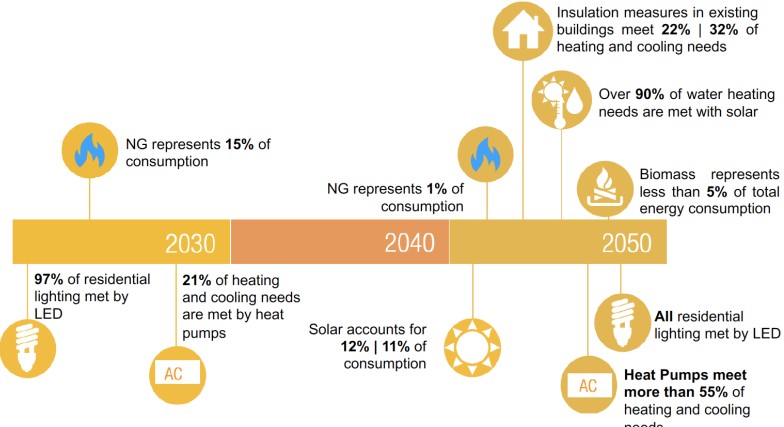

**Figure 3.** Timeline of residential sector's carbon neutrality pathway up to 2050, as defined by the RNC2050, adapted with permission from Ref. [33].

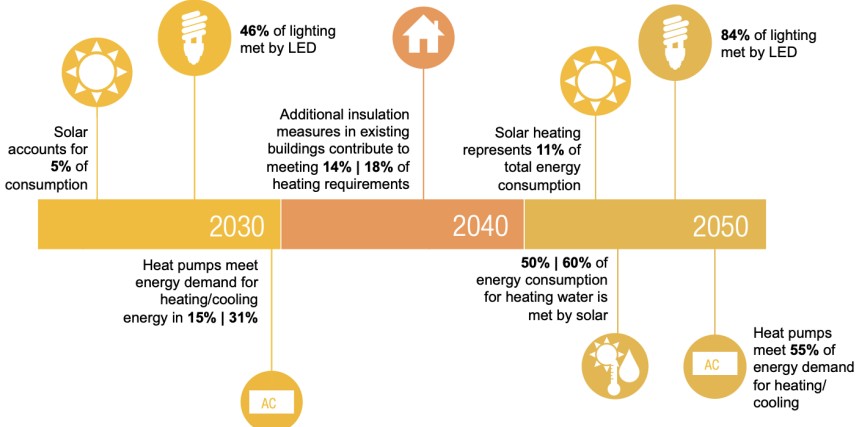

**Figure 4.** Timeline of service sector's carbon neutrality pathway up to 2050, as defined by the RNC2050, adapted with permission from Ref. [33].

## 3. Local Energy Planning Assistant—LEPA

The LEPA software [34] is a tool for the decision-making process of local energy planning. This tool can incorporate the values and preferences of the local actors into the energy planning process to maintain transparency and create well-balanced decisions. The tool consists of an MS Excel spreadsheet including several sheets developed in [34]. This tool is divided into two primary sections: (i) The modelling of the energy system for the base year and respective evolution over time, the reference scenario; and (ii) The creation and evaluation of alternatives, which will be explained in the following sub-chapters. The reference scenario is the starting point for developing and evaluating alternative pathways, i.e., the alternatives are compared against the reference scenario. Figure 5 presents the framework applied in this work through LEPA, organised into four sections: start, inputs, methodology and results.

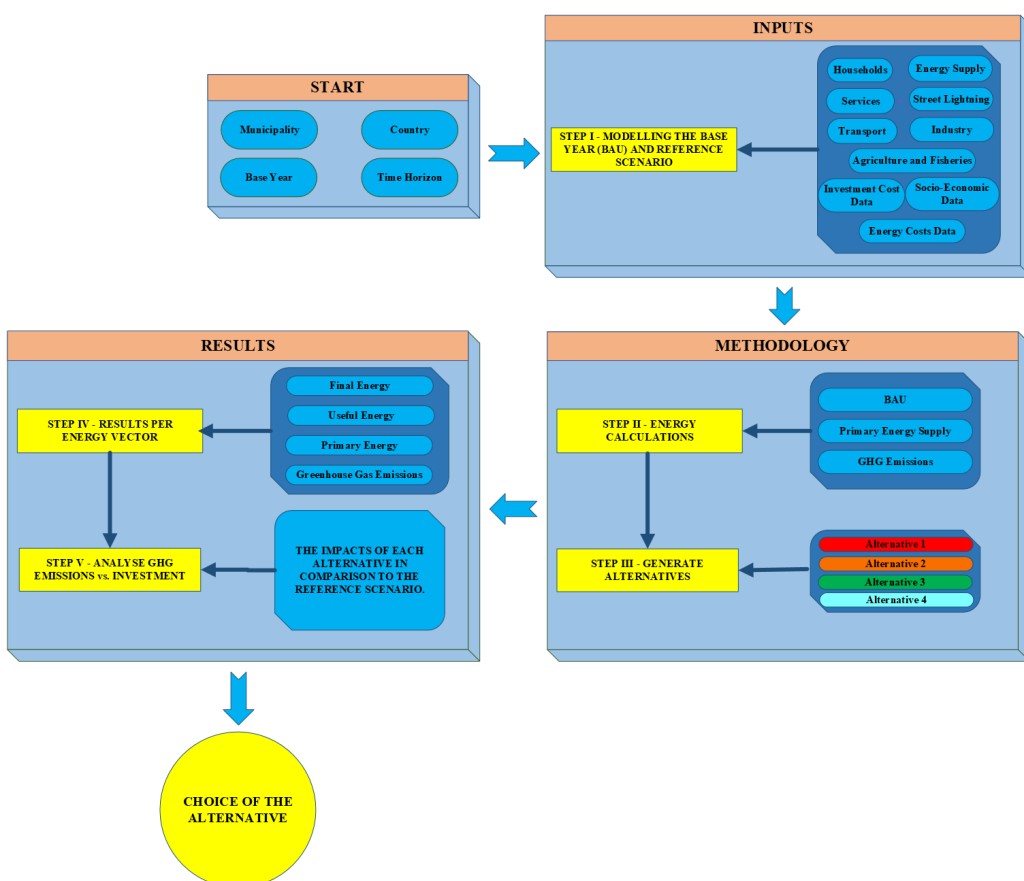

**Figure 5.** Flowchart of the functioning of the Local Energy Planning Assistant tool.

### 3.1. Start

The "Start" Section 3.1 sheet is used to specify the location (municipality/region and country), the base year and the time horizon for which the target was set, and links to all the other sheets. Note that in the present work, the territorial scope of this tool is the North of Portugal. Additionally, it was determined that 2020 was the base year and 2050 was the time horizon for energy policies' assessment.

### 3.2. Inputs

Section 3.2 is one of the main sections, being crucial for the tool implementation. It consists of only one step (Figure 5), which is the modelling of the energy system for the base year and building the reference scenario. It involves filling input sheets concerning the energy demand and supply for the base year. Finally, it is necessary to enter the data related to energy costs, investment-cost and socio-economic indicators for the reference

year as well as the expected evolution over time. The projection of the evolution of the energy system within the time horizon is performed based on the expected evolution of the socio-economic indicators (population, transportation habits, economic growth, etc.), as well as technological evolution (average efficiency, etc.).

*3.3. Methodology*

This sub-chapter is divided into two main steps (steps 2 and 3) and concerns the methodology used in the tool for energy planning. The second step illustrated in Figure 5 is dedicated to the characterisation of the Business As Usual (BAU) scenario in terms of primary energy supply, final energy demand and GHG emissions. The BAU refers to the natural evolution of the energy system, being a possible comparison to the alternatives. The primary energy supply refers to the energy derived from natural resources such as coal, crude oil and natural gas. These calculations are based on the final energy demand for the selected area/region, in the case of this work, the North of Portugal. It also considers the national energy balance for the characterisation of the energy transformation processes. They represent essential data for the remaining sheets of the tool, namely the Electricity Mix, in which the evolution of GHG emissions is calculated. Finally, GHG emissions were determined based on the objectives defined in the RNC [33], where their development until 2050 is presented. Through a complete analysis of this document, it is known that in 2050 there should be a total decarbonisation of the electricity generation sector, namely 100% renewables in electricity production. It is also known that the consumption of coal will end in 2030. Concerning RES, it is possible to understand through the roadmap that photovoltaic technology will assert itself with more significant evidence, as well as wind energy which increases its participation significantly. These two technologies will jointly ensure 50% of the electricity generated in 2030 and 70% in 2050, respectively. It is also intended to reach 94% in 2030, 97% in 2040 and 100% of renewables in electricity generation in 2050 [33]. This information in the RNC [33] made it possible to determine what percentages to put in the Electricity Mix sheet for the portion of electricity.

The third step illustrated in Figure 5 refers to the development of different alternatives to the reference scenario. Using the alternatives generation table included in the LEPA tool, it is possible to generate options by combining activities with the appropriate degree of implementation. For example, it is possible to apply measures related to thermal insulation, space heating and cooling, appliance efficiency, and lighting efficiency, among others.

*3.4. Results*

The analysis of the results obtained through the LEPA tool is divided into two main steps. The fourth step (shown in Figure 5) concerns the analysis of the results by energy vector, namely final energy, useful energy, primary energy and GHG emissions.

Consequently, the fifth step (depicted in Figure 5) concerns the analysis of reducing GHG emissions vs. the investment cost for its implementation and the respective impacts of each alternative compared to the reference scenario. In this way, it is possible to determine the impact in terms of GHG emissions and the required investment. The decision-makers can better understand the balance between the expected investment and the benefits of each choice. Thus, it is possible to select the best alternative after completing the five steps mentioned above.

## 4. Assessment of Energy Policies in The North of Portugal

This section presents the application of the LEPA tool to assess current energy policies in the context of Northern Portugal. In addition, it presents the results of potential measures (alternatives) that can be adopted to achieve carbon neutrality by 2050, considering a significant reduction in the expected investment costs. A comparison of the different alternatives is also provided.

### 4.1. Case Characterisation

The North of Portugal has approximately 3.6 million inhabitants, which corresponds to 35% of the total population of Portugal. This region consists of 86 municipalities and 1426 parishes [35]. In addition, the municipalities are organised into eight inter-municipal communities, which constitute level III of the Nomenclature of Territorial Units for Statistics (NUTS III), approved by the European Commission. The eight sub-regions that make up the North of Portugal are as follows [35]: "Alto Tâmega", "Alto Minho", "Área Metropolitana do Porto", "Ave", "Cávado", "Douro", "Tâmega e Sousa", and "Terras de Trás-os-Montes".

Obtaining the input data is the first stage when using the end-use energy model. Because not all the necessary information is readily available at the municipal level, this is undoubtedly one of the most difficult tasks when applying energy planning models. Numerous assumptions and scaling-down from other administrative levels, such as regional or national, have been made due to data availability restrictions and a lack of bottom-up municipal data. For the residential sector, the estimates were based on the number of occupied residential buildings in the North of Portugal taken from [36] until 2020 and on the energy balance for 2020 [37]. On the other hand, for the service sector, the estimates were determined based on the number of employed population for 2020 in the North of Portugal [38] and also on the energy balance for 2020 [37].

### 4.2. Methodology of the Alternatives

This section concerns the methodology behind the Alternatives and the rationale behind them (Table 1). It should be noted that the use of the term Alternatives refers to EPA with different characteristics. Alternative 1 is based on the PNEC. By applying this alternative, whose objectives are directed to 2030, it is possible to have a closer view of the results to be achieved since they are projections for eight years from now. It was determined that Alternative 2 would be based on the RNC, which sets the achievement of carbon neutrality by 2050 as a commitment. On the other hand, Alternative 3 is a new one consisting of implementing measures related to the replacement of less efficient appliances for more efficient ones. That is, it aims to improve EE in buildings through the refurbishment of all appliances in buildings. Finally, Alternative 4 consists of combining the measures implemented in Alternatives 2 and 3 to reach a GHG value in 2050 close to zero without drastically increasing the value of the investment cost.

Table 1 summarises the level of implementation of the different types of use in each alternative (+, ++ or +++). Each category represents a level of implementation for a given type of use, allowing for a comparison between alternatives. It is important to point out that Table 1 does not quantify the share of measures adopted in each type of use but rather correlates to their impact in each alternative. For instance, measures toward water heating have a higher relative importance in Alternative 2 than in Alternative 4. Therefore, it is possible to verify that Alternatives 2 and 4 are the most complete choices and, consequently, the ones that obtain the most satisfactory results. It should be considered that n/a means not applicable.

**Table 1.** Analysis and comparison of the measures applied in the different types of use for each alternative.

|  | Alternative 1 | Alternative 2 | Alternative 3 | Alternative 4 |
|---|---|---|---|---|
| Water Heating | n/a | +++ | n/a | ++ |
| Thermal Insulation | +++ | ++ | n/a | ++ |
| Space Heating and Cooling | ++ | +++ | n/a | ++ |
| Energy Efficiency | ++ | n/a | ++ | + |
| Household Appliances | n/a | n/a | +++ | ++ |
| Lightning | n/a | +++ | n/a | n/a |
| Renewable Energies | ++ | ++ | n/a | n/a |
| Cooking | n/a | + | n/a | n/a |
| GHG Emissions | ++ | ++ | + | +++ |

### 4.3. Results

This section exposes the main results of the reference scenario (BAU) and the simulations for the selected alternatives, taking into account their impact on GHG emissions and the investment cost for their deployment. For the sake of simplicity, it is noteworthy that special emphasis is given to Alternatives 2 and 4 due to their characteristics to achieve carbon neutrality by 2050. A comparison of the alternatives is also highlighted.

#### 4.3.1. Business-As-Usual Scenario

The business-as-usual (BAU) scenario consists of simulating all policies, actions and measures currently in force. The time horizon established for analysing the results is from 2020 to 2050. An analysis every 10 years was determined to simplify the perception of the evolution of the outcomes. In Figures 6 and 7, it is possible to observe the evolution of the GHG emissions for the North of Portugal, according to BAU for the residential and service sector, respectively. The values obtained through the reference scenario will be compared with the results obtained in each alternative. As expected, it is possible to conclude that the values obtained for the BAU are higher than the alternatives, both for the residential and the service sector.

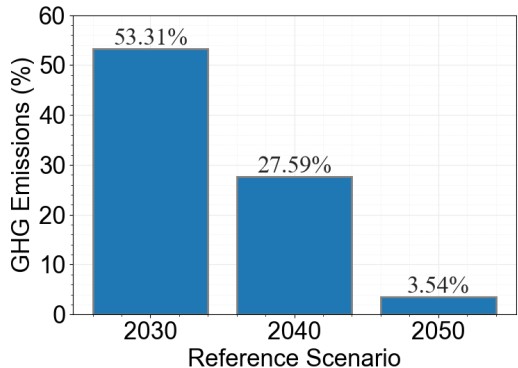

**Figure 6.** BAU—Evolution of GHG emissions for the residential sector (%).

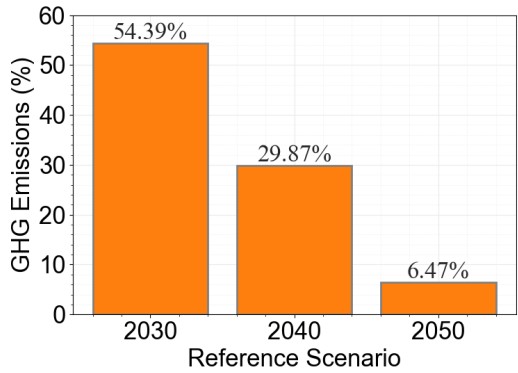

**Figure 7.** BAU—Evolution of GHG emissions for the service sector (%).

#### 4.3.2. Alternative 1

This alternative is based on PNEC, which has set goals for 2030, providing a clearer view of anticipated results as projections are made for the near future. PNEC's strategic goals include encouraging energy transition by committing to renewable energies and EE, as well as integrating mitigation goals into sectorial policies. These objectives will help achieve carbon neutrality by 2050.

Taking into consideration that this work is focused on residential and service buildings, targets for RES in transport and electricity interconnections were not considered. Figure 8

presents the comparison between emissions and investment costs until 2050. In 2030, the $CO_2$ emission reduction was estimated at 76.7% in 2030 and reached 98.7% in 2050. The analysis performed for the residential sector, as demonstrated in Figure 9, has a similar trend in terms of emissions reduction. In 2030, it was projected an 81.9% $CO_2$ reduction, reaching 98.1% by 2050.

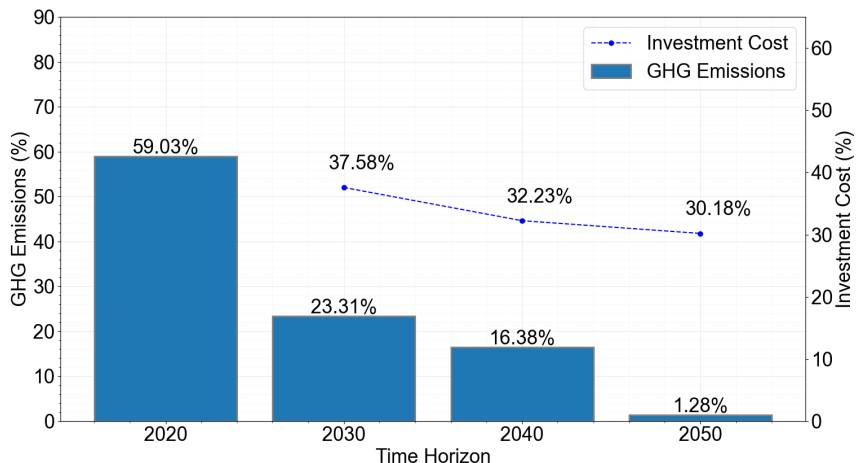

**Figure 8.** Alternative 1—Comparison between emissions and the investment cost for the residential sector.

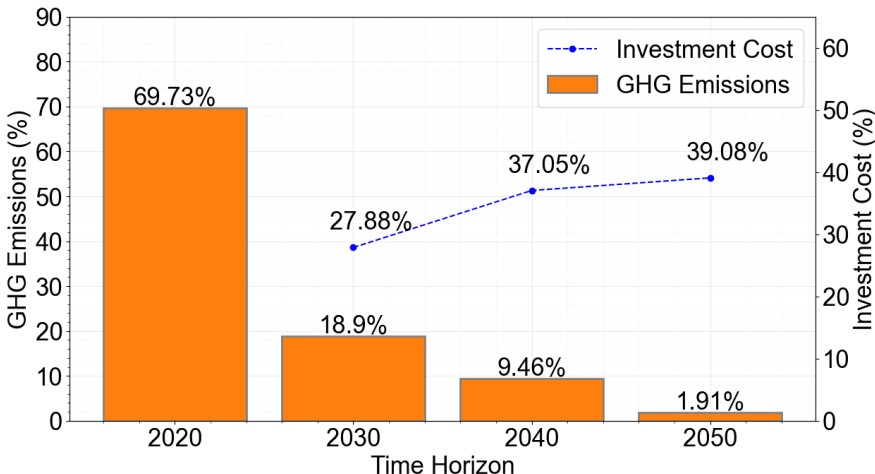

**Figure 9.** Alternative 1—Comparison between emissions and the investment cost for the service sector.

### 4.3.3. Alternative 2

In buildings, energy is used to provide energy services such as space heating and cooling, lighting, refrigeration and food preparation, and sanitary water heating, among others. It is known that the building sector is responsible for 5% of the country's GHG emissions [33]. Alternative two is based on RNC targets detailed in Section 2.2.

Figure 10 presents the comparison between the evolution of GHG emissions and the investment cost after the implementation of proposed measures for the residential sector for Alternative 2. Emissions were forecasted to decrease from 68.26% in 2020 to 1.65% in 2050. In terms of investment cost, a consistent reduction is also observed. In 2030, costs are predicted to be at 57.35%, considerably reducing to 9.79% by 2050. As it is possible to observe, Alternative 2 allows an emission reduction of about 98.35% when compared to 2020 for the residential sector.

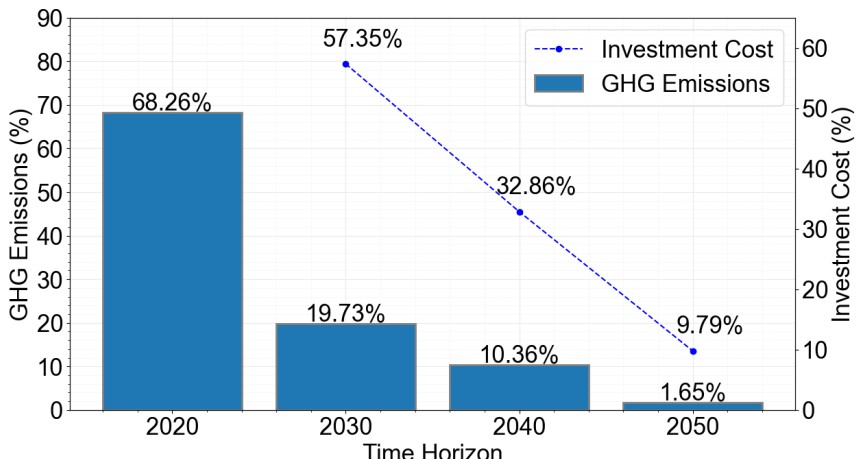

**Figure 10.** Alternative 2—Comparison between emissions and the investment cost for the residential sector.

For the service sector, the principle for identifying the measures to be implemented for Alternative 2 is similar to the one used for residential buildings. Hence, Figure 11 presents the comparison of the evolution of GHG emissions and the evolution of investment costs over the years for the service sector. For this sector, it is possible to verify that GHG emissions are reduced by approximately 97.1%, compared with the 2020 values. The increased investment refers to the growing investment cost of implementing space heating and lighting technologies.

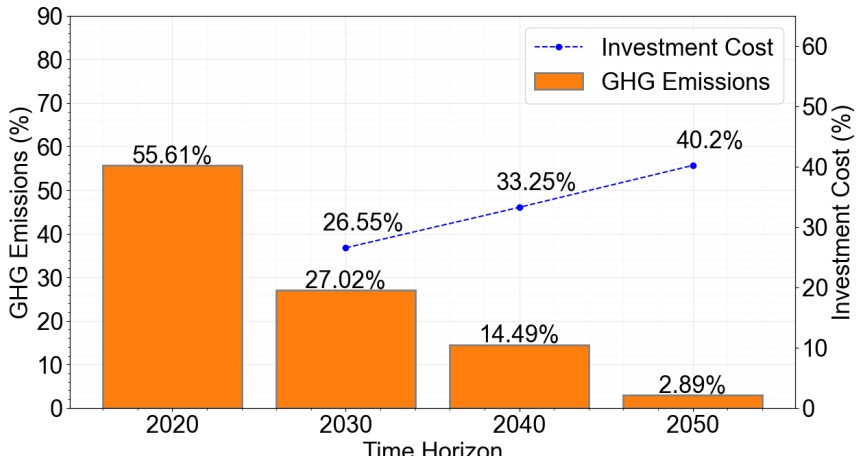

**Figure 11.** Alternative 2—Comparison between emissions and the investment cost for the Service Sector.

### 4.3.4. Alternative 3

This alternative was formulated to enhance EE in household appliances and complement the existing plans, namely the PNEC and RNC. Both plans lack specific measures for replacing less efficient appliances with more efficient ones. The results of Alternative 3 indicate that investing in appliances can yield similar cost benefits to replace space heating and hot water equipment, as presented in Alternatives 1 and 2. Figures 12 and 13 present the comparison between emissions and investments by 2050. The residential sector presented a GHG emission of approximately 71.2% in 2030 and reached 98.1% in 2050. With a similar pattern, the service sector presented a $CO_2$ emission reduction of 71.5% and 66.9% for 2030 and 2050, respectively.

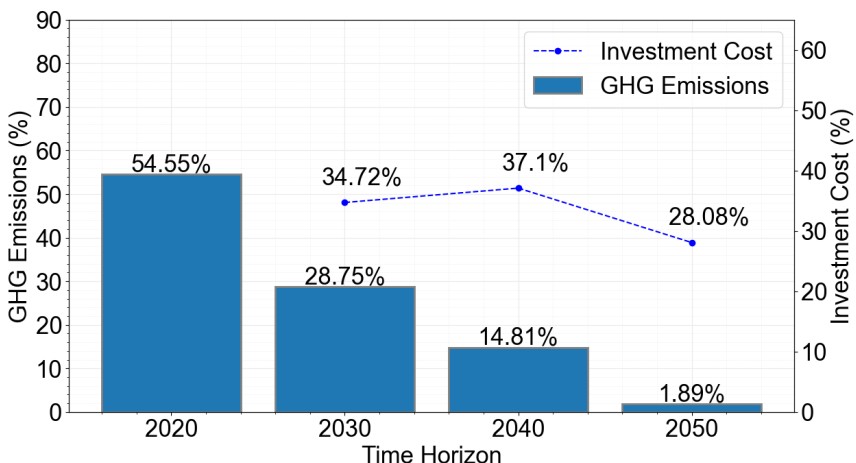

**Figure 12.** Alternative 3—Comparison between emissions and the investment cost for the Residential sector.

Considering that GHG emissions did not decrease significantly when compared to the previous alternatives, we propose a fourth alternative. This one integrates the measures of Alternative 3 with strategic measures for water and space heating. This alternative aims to create synergies between the strengths of Alternatives 2 and 3 to obtain a new and more efficient techno-economic solution.

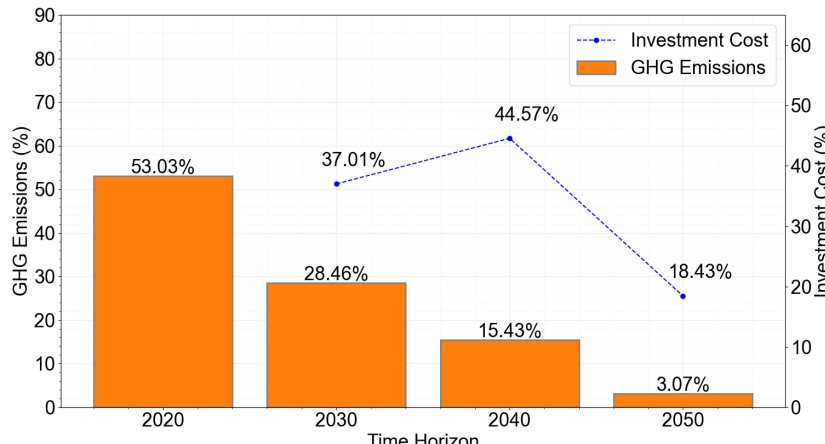

**Figure 13.** Alternative 3—Comparison between emissions and the investment cost for the Service sector.

### 4.3.5. Alternative 4

This alternative is based on the measures implemented in Alternatives 2 and 3. It is verified that, both in the PNEC and the RNC, there are no specific measures for the replacement of less efficient appliances by more efficient ones. Therefore, Alternative 4 is a solution to improve the results obtained for GHG emissions with lower investment costs. Adjusting the values initially defined for each measure was necessary to obtain better results. The results obtained from the combination of these two alternatives are shown in this section.

For the residential sector, the comparison between the evolution of GHG emissions and the investment cost was determined, as depicted in Figure 14. It is possible to verify that, with this alternative, the emissions present a reduction of nearly 99%, compared with the 2020 values. The investment cost of these measures follows this decrease, verifying that this alternative would be a good solution to implement.

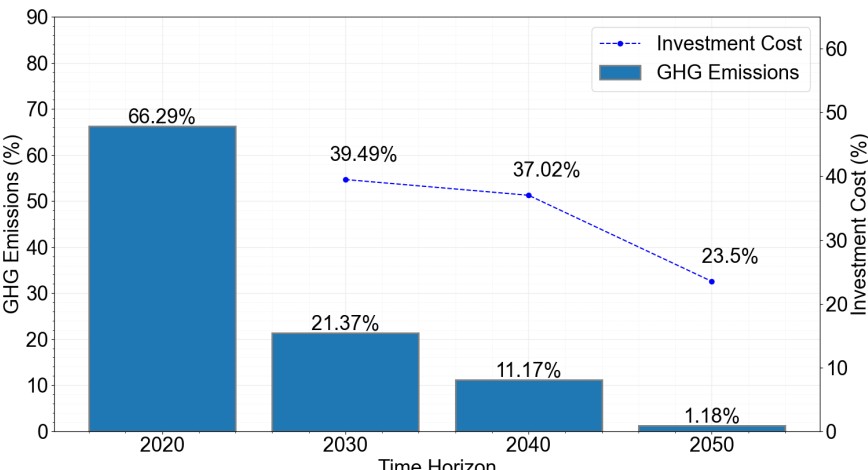

**Figure 14.** Alternative 4—Comparison between emissions and the investment cost for the residential sector.

The analysis of the results obtained for the service sector is similar to that for the residential sector. In this case, the measure implemented in Alternative 3 for this sector remains unchanged, and then the measures of Alternative 2 that influence the decrease of GHG emissions are added. Finally, Figure 15 presents a comparison of the evolution of emissions with the investment cost. Once again, it can be seen that the emission values will be close to zero in 2050 and that the investment cost of these measures follows this trend.

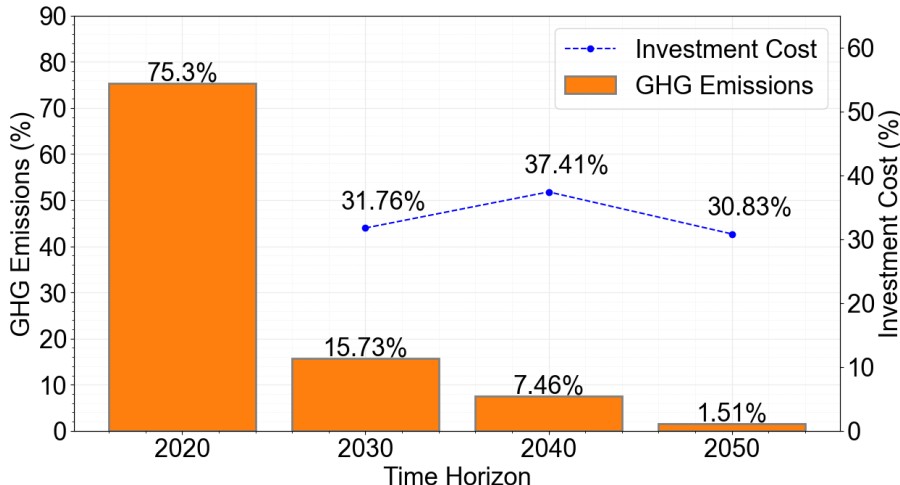

**Figure 15.** Alternative 4—Comparison between emissions and the investment cost for the Service Sector.

### 4.3.6. Comparison between Alternatives

This section is designed to simplify the comparison and the discussion of all alternatives. Figures 16 and 17 present the evolution of emissions and the investment cost of each alternative for the residential buildings sector, respectively. A similar analysis is illustrated in Figures 18 and 19 for the service building sector.

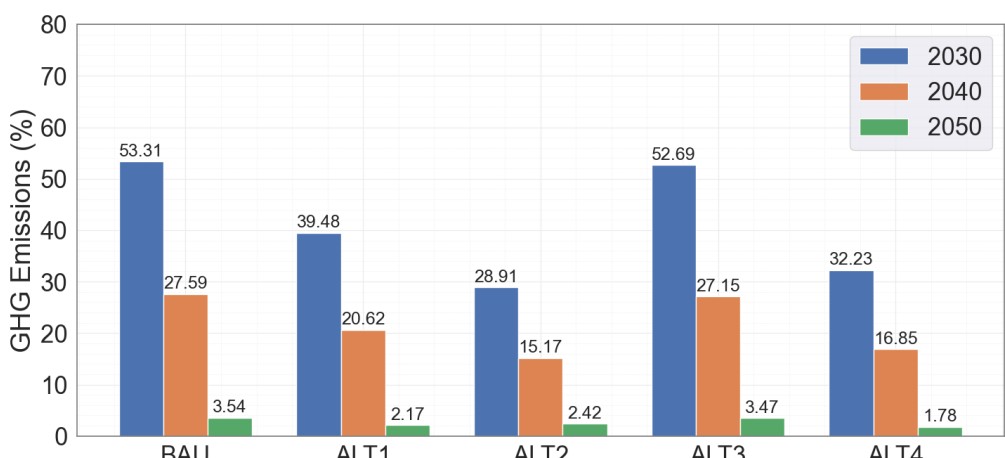

**Figure 16.** Comparison of the evolution of GHG emissions between alternatives for the residential sector (%).

By analysing Figure 16 concerning the residential sector, although Alternative 2 is worthwhile in the short term, it is possible to verify that Alternative 4, which combines measures implemented in Alternative 2 and Alternative 3, presents better results in terms of emissions. A reduction of about 98% of GHG emissions between 2020 and 2050 is achieved. Regarding the required investment cost (illustrated in Figure 17), one can derive different conclusions depending on the perspective of the analysis. It should be mentioned that the total investment throughout the whole time frame gives different directions when compared with single-time analysis. That is, some alternatives may induce a higher investment cost in the beginning than close to the time horizon or vice versa. It is possible to verify that Alternatives 3 and 4 are the ones that require the least investment, with a total investment of approximately 4738 and 4567 M€, respectively. Alternative 4, becomes a better solution than Alternative 3 because it is more complete in terms of the application of measures in other energy vectors and due to its GHG emissions value. With the creation of Alternative 4, an attempt was made without radically increasing the value of the investment cost of Alternative 2 to reach GHG emissions levels close to zero in 2050. One can conclude that this objective was accomplished, considering Alternative 4 a good option to implement.

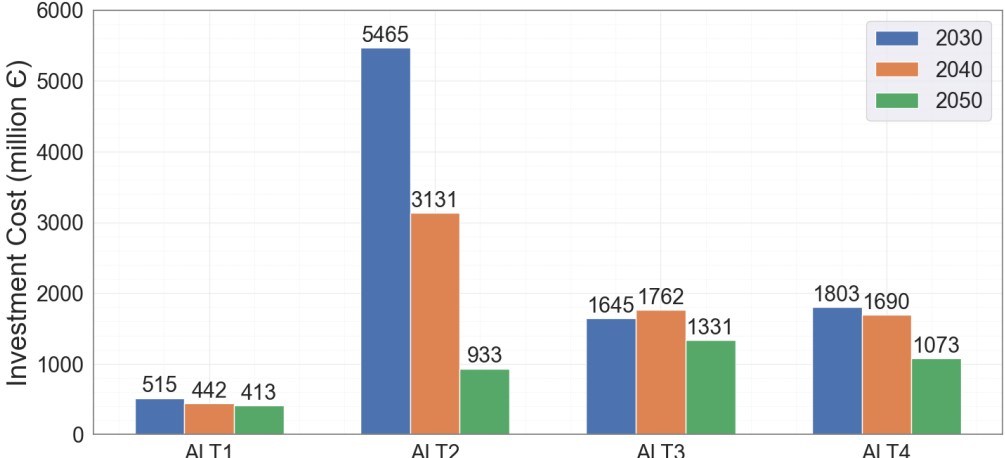

**Figure 17.** Comparison of the evolution of investment costs between alternatives for the residential sector (million €).

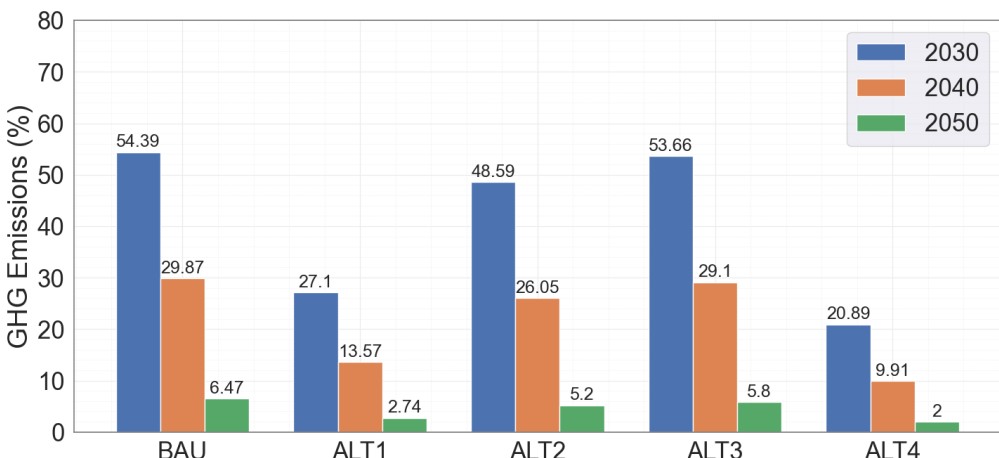

**Figure 18.** Comparison of the evolution of GHG emissions between alternatives for the service sector (%).

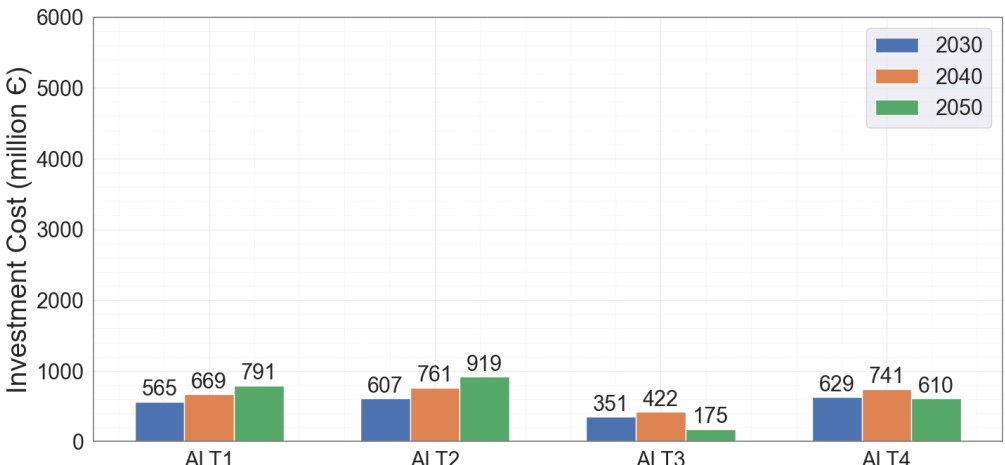

**Figure 19.** Comparison of the evolution of investment cost between alternatives for the service sector (million €).

For the service sector, a similar analysis was carried out. In Figure 18, concerning GHG emissions, it is verified that Alternative 4 is, in fact, a good alternative in comparison to the reference scenario. Once again, one can observe a significant reduction in the value of GHG emissions in 2050, when compared to 2020, of about 98%. Regarding the investment cost (illustrated in Figure 19), one can verify that Alternative 3 is the one that implies a lower investment, around 948 M€. However, this alternative concerns only measures related to the replacement of appliances by more energy-efficient appliances. Alternative 4, compared to Alternative 2, presents a better result since it implies a lower total investment cost, around 1980 M€ and presents a GHG emissions value close to zero in 2050.

### 4.3.7. Discussion of the Results

The four alternatives presented in this study focus on reducing GHG emissions and achieving full decarbonisation of the energy system, yet through different actions. Alternative 1 prioritises thermal insulation and space heating and cooling measures, while Alternative 2 concentrates on water heating and lighting, with additional measures related to space heating and cooling. In contrast, Alternative 3 only targets the replacement of household appliances. Alternative 4 combines the strengths of Alternatives 2 and 3 to deliver more effective GHG emissions reduction with lower investment costs. The analysis reveals that Alternative 4 offers significant investment cost savings of around 52% for the residential sector and 13% for the service sector compared to Alternative 2. At the same

time, it also provides GHG emission reduction gains of 0.64% for the residential sector and 3.2% for the service sector in 2050 compared to Alternative 2. The findings offer valuable insights for researchers, practitioners, and policymakers, as the study provides more precise and accurate information for developing and modelling effective energy policies. The measures proposed in the alternatives also have multiple benefits for consumers, enabling them to reduce energy costs while actively contributing to carbon neutrality by 2050.

The study highlights the importance of building energy retrofitting to achieve carbon neutrality and improve energy efficiency. A comprehensive intervention in the building's various elements, such as insulation, the use of more efficient air conditioning and hot water production equipment, and renewable energy systems like solar thermal or photovoltaic systems, can enhance the building's energy efficiency, improve comfort conditions, and reduce energy bills. Such actions are necessary to meet the national and European energy and climate action plans' objectives and targets. Therefore, promoting energy efficiency in residential and service areas should be a priority investment.

However, the authors acknowledge that this study has limitations that should be taken into consideration by potential stakeholders. Namely, the scope of work is delimited by the characteristics of the energy systems in the north of Portugal. Thus, the results can be replicated in other national regions with similar characteristics, to some extent. However, it should be mentioned that the application of this framework to different regions within Portugal or other countries would require an appropriate assessment and adjustments of the input parameters used in this model. Another important limitation of the model is that, for a proper and accurate assessment of local building energy planning, it is necessary to access accurate data from buildings and energy systems, such as the current energy characteristics of buildings and local energy systems.

Bearing this in mind, the authors identified a few points for further research on this subject, aiming for improved applicability and generalization of the tool. More precisely, we propose the following ideas for continuous development of the LEPA tool: (i) Include an analysis of batteries in the short/medium term, as they make it possible to support the need to balance supply and demand and transfer electricity from one period to another, increasing energy self-sufficiency; (ii) continue updating the timeline of EE policies in buildings in Portugal and carry out this study for renewable energies and the introduction of electric vehicles, to simplify information for citizens; and (iii) extend this case study to the Central and Southern Portugal, as well as Archipelagos, as they may have different energy policy needs, depending on their current situation and advancements of EE in buildings.

## 5. Conclusions

The challenges towards carbon neutrality are huge and should be promoters of structural changes aimed at economic growth and improving the quality of life of citizens. An important part of energy consumption is through buildings, so the complete decarbonisation of buildings is a must. To this end, new energy policies must be designed and developed, accounting for the reality of the different environments.

This work aimed to develop an alternative energy policy that achieves near-zero GHG emissions by 2050, focused on the North of Portugal whilst not leading to larger investment requirements when compared to the current EPA. The proposed alternative showed to be more efficient, achieving a reduction in GHG emissions (about 0.64%) with a significantly lower investment cost (about 52%) for residential buildings.

An important conclusion is that even though the energy policies in force in Portugal aim at the complete decarbonisation of the buildings sector, there are still several possible and cost-effective ways to achieve the desired targets with lower investment costs.

**Author Contributions:** Conceptualization, S.C., T.S., I.A., W.F. and M.A.M.; Data curation, S.C., T.S. and I.A.; Formal analysis, S.C., T.S., I.A. and W.F.; Investigation, S.C. and T.S.; Methodology, S.C, T.S. and I.A.; Supervision, T.S., W.F. and M.A.M.; Validation, S.C. and T.S.; Writing—original draft, S.C. and T.S.; Writing—review & editing, S.C., T.S., I.A., W.F. and M.A.M. All authors have read and agreed to the published version of the manuscript.

**Funding:** This research received partial support from the Norte Portugal Regional Operational Programme (NORTE 2020), under the PORTUGAL 2020 Partnership Agreement, through the European Regional Development Fund (ERDF), within the DECARBONIZE project under agreement NORTE-01-0145-FEDER-000065 and by the Scientific Employment Stimulus Programme from the Fundação para a Ciência e a Tecnologia (FCT) under the agreement 2021.01353.CEECIND.

**Data Availability Statement:** Not applicable.

**Conflicts of Interest:** The authors declare no conflict of interest.

## Abbreviations

The following abbreviations are used in this manuscript:

| | |
|---|---|
| BAU | Business As Usual |
| DL | Decree-Law |
| ECS | Energy Certification System |
| EE | Energy Efficiency |
| EnC | Energy Community |
| EPA | Energy Policies and Actions |
| EPBD | Energy Performance of Buildings Directive |
| EU | European Union |
| GHG | Greenhouse Gas |
| LEPA | Local Energy Planning Assistant |
| NZEB | Near Zero Energy Building |
| PNEC | National Energy and Climate Plan |
| RES | Renewable Energy Sources |
| RNC | National Carbon RoadMap |

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
