# Peer review of "Design of an Energy Policy for the Decarbonisation of Residential and Service Buildings in Northern Portugal"

_energies, doi:10.3390/en16052239_

Round 1
Reviewer 1 Report
The article ‘’ Analysis of energy policies for the decarbonisation of energy consumption in buildings: The case of the Northern region of Portugal’’ is interested article, however, it require major changes. Moreover the technical, novel side of the paper is very week. Following are the comments for the authors to improve the article:
Ø The article title is a general statement, authors are suggested to re-write the title with the research impact and novelty perspective.
Ø Avoid using words like ‘’ Northern region of Portugal’’.
Ø A lot of research is going on decarbonisation of energy consumption in buildings. How authors claim the new aspect for the need of this publication?
Ø Problem statement should be mentioned at the start of the abstract, it is totally missing.
Ø There should be some statistical figured values in the abstract which can quantify the research / optimization and it can make readership of the journal easy.
Ø Abstract needs to re-write as it is not clear and number of abstract components are missing.
Ø There should be some proper synchronization of the sentences in meaningful way.
Ø The abstract should also include the solution of the problem based on the problem statement with some particular application/s.
Ø Formatting and style of all tables must be same.
Ø 2. Portuguese Energy Policies should be the part of introduction section. There is no need for new section.
Ø Authors must avoid sub para in introduction section.
Ø Line no. 162-166: authors should avoid short paras and this should be checked throughout the paper.
Ø The quality of figures is very poor. All figures require re-work.
Ø Figure 2 text is not visible. Should be redrawn.
Ø Figure 4 is from literature. It should not be the part of author’s results section.
Ø There are few old references, authors are encouraged to add latest literature.
Ø The graphics of the results need improvement. It should be reviewed by the authors in the revised version.
Ø Authors should only focus on the conclusion of the research and not to add results and/or any other information. The section length is too long.
Author Response
Dear reviewer,
We thank you for all the suggestions to improve our work. All of them were taken into account. Please have a look at the attached file which has detailed answers to the comments.
Kind regards,
The Authors

Author Response

(The authors gave the same response as above.)

Reviewer 3 Report
The article is well conceptualized, and the problem is explained numerically and graphically with the help of sufficient analysis. The English language of the article is good, but the authors should carefully re-check the syntax or grammatical errors. Authors should check if they follow the suggested style of references in text, and tables as well in the bibliographic list.
I have the following suggestions/queries for further quality improvement of the manuscript:
1. Both the length and the content of the abstract seem appropriate.
2. Keywords should be chosen that reflect the essence of the work and are distinct from the title.
3. The Results section contains only the results of the present study. However, the discussion section should be added to discuss the results obtained in the case of similarities and differences and support them with further details and justifications.
4. Limitations are not included in the present form.
5. Future directions are not clearly stated.
6. Conclusions should be thoroughly revised. They should be clear and concise. Some sentences are redundant. Summarize the most important points.
Author Response

(The authors gave the same response as above.)

Round 2
Reviewer 1 Report
Accept